# Technical note: Uncertainties in eddy covariance $CO_2$ fluxes in a semi-arid sagebrush ecosystem caused by gap-filling approaches

Jingyu Yao[1], Zhongming Gao[2], Jianping Huang[1], Heping Liu[2], and Guoyin Wang[3]

[1]Key Laboratory for Semi-Arid Climate Change of the Ministry of Education, College of Atmospheric Sciences, Lanzhou University, Lanzhou, China

[2]Laboratory for Atmospheric Research, Department of Civil and Environmental Engineering, Washington State University, Pullman, Washington, USA

[3]Department of Atmospheric and Oceanic Sciences & Institute of Atmospheric Sciences, Fudan University, Shanghai, China

*Correspondence to*: Jianping Huang (hjp@lzu.edu.cn)

**Abstract.** Gap-filling eddy covariance $CO_2$ fluxes is challenging at dryland sites due to small $CO_2$ fluxes. Here, four machine learning (ML) algorithms including artificial neural network (ANN), k-nearest neighbours (KNN), random forest (RF), and support vector machine (SVM) are employed and evaluated for gap-filling $CO_2$ fluxes over a semi-arid sagebrush ecosystem with different lengths of artificial gaps. The ANN and RF algorithms outperform the KNN and SVM in filling gaps ranging from hours to days, with the RF being more time efficient than the ANN. Performances of the ANN and RF are largely degraded for extremely long gaps of two months. In addition, our results suggest that there is no need to fill the daytime and nighttime NEE gaps separately when using the ANN and RF. With the ANN and RF, the gap-filling induced uncertainties in the annual NEE at this site are estimated to be within 16 g C m$^{-2}$, whereas the uncertainties by the KNN and SVM can be as large as 27 g C m$^{-2}$. To better fill extremely long gaps of a few months, we test a two-layer gap-filling framework based on the RF. With this framework, the model performance is improved significantly, especially for the nighttime data. Therefore, this approach provides an alternative in filling extremely long gaps to characterize annual carbon budgets and interannual variability in dryland ecosystems.

## 1 Introduction

The eddy covariance (EC) technique has been widely applied for monitoring energy and water fluxes as well as net ecosystem exchanges of carbon dioxide (NEE) and other trace gases between lands and the atmosphere (Baldocchi, 2003; Oncley et al., 2007; Stoy et al., 2013). However, due to multiple factors including power outages, instrument malfunctions and maintenance, and data quality checks, there exist gaps with approximately 20-60% of half-hourly data points annually at many long-term EC sites (Dragoni et al., 2007; Falge et al., 2001; Ma et al., 2007; Missik et al., 2019, 2021; Moffat et al., 2007; Pastorello et

al., 2020; Soloway et al., 2017; Wutzler et al., 2018). An average gap fraction of 30% in an annual dataset leads to an uncertainty of $\pm 25$ g C m$^{-2}$ year$^{-1}$ for the annual NEE at forest sites (Moffat et al., 2007), while some EC sites report much greater uncertainties (Soloway et al., 2017). Therefore, gap-filling usually accounts for one large source of uncertainties in the annual NEE (Soloway et al., 2017), together with other sources of uncertainties such as measurement errors and bias related to non-closure of the surface energy balance (Gao et al., 2019; Wilson et al., 2002).

Robust NEE gap-filling approaches are critical for quantifying the annual and interannual variability of carbon budgets (Falge et al., 2001; Irvin et al., 2021; Moffat et al., 2007; Pastorello et al., 2020; Richardson and Hollinger, 2007; Soloway et al., 2017; Wutzler et al., 2018). Previous studies have developed and evaluated a number of NEE gap-filling approaches including non-linear regressions (NLR), look-up tables (e.g., marginal distribution sampling (MDS)), machine learning (ML) algorithms (e.g., artificial neural networks), and process-based models (Falge et al., 2001; Huang and Hsieh, 2020; Moffat et al., 2007; Reichstein et al., 2005; Wutzler et al., 2018). NLR fills NEE gaps based on regression analyses between NEE and meteorological variables such as temperature (e.g., air or soil temperature) and light (e.g., photosynthetically active radiation), whereas MDS is based on look-up tables for similar meteorological conditions (i.e., global radiation, air temperature, and vapor pressure deficit) (Falge et al., 2001; Moffat et al., 2007; Reichstein et al., 2005). By virtue of an easy-to-use R package (Wutzler et al., 2018), MDS has become the standard method for NEE gap-filling (e.g., Pastorello et al., 2020), although it cannot effectively fill the gaps of longer than 12 days (Moffat et al., 2007). ML-based methods are trained by presenting them with numerous meteorological variables as inputs and NEE as output data, which have the potential to fill long gaps (Dengel et al., 2013; Kim et al., 2020; Moffat et al., 2007). Artificial neural network (ANN), for instance, has been widely used for gap-filling $CO_2$ and $CH_4$ fluxes across a variety of EC sites at forests, grasslands, croplands, and wetlands (Dengel et al., 2013; Delwiche et al., 2021; Huang and Hsieh, 2020; Irvin et al., 2021; Kim et al., 2020; Knox et al., 2016; Mahabbati et al., 2021; Moffat et al., 2007; Papale and Valentini, 2003; Soloway et al., 2017). More recently, other ML algorithms such as random forest (RF), k-nearest neighbours (KNN), and support vector machine (SVM) have recently been assessed for flux gap-filling over different ecosystems, and RF is found to be outperformed the other ML algorithms as well as the MDS method (Huang and Hsieh, 2020; Irvin et al., 2021; Kim et al., 2020; Mahabbati et al., 2021). However, the performance of these ML-based algorithms has not been evaluated in filling gaps in EC fluxes for dryland ecosystems with low NEE.

Besides the selection of gap-filling algorithms, several other factors may also degrade the performance of the algorithms and cause uncertainties in gap-filled fluxes. For example, the performance of gap-filling algorithms decreases with increasing the gap length (Huang and Hsieh, 2020; Irvin et al., 2021; Kim et al., 2020), and thus long gaps in $CO_2$ flux are considered one of the primary uncertainty sources of NEE estimation (Aubinet at al., 2012). In addition, spatial variability of $CO_2$ flux and meteorological drivers (e.g., soil temperature) due to heterogeneous landscapes around flux towers (Chu et al., 2021; Stoy et al., 2013) can lead to unknown bias in modelling research (Metzger, 2018). That is, the trained ML algorithms using the measured $CO_2$ flux and meteorological variables may not well reflect their real relationship within flux footprints and induce bias to gap-filled fluxes and the annual NEE.

Dryland ecosystems, comprising around 40% of the Earth's land surface, play a critical role in determining the trend and interannual variability of the global terrestrial carbon budgets (Ahlström et al., 2015; Missik et al., 2021; Yao et al., 2020), though the expansion of projected global drylands under climate change is still under debate (Berg and McColl, 2021; Feng
and Fu, 2013; Huang et al., 2015; Yao et al., 2020). Long-term continuous measurements of land surface fluxes over dryland ecosystems are critical for assessing the impact of climate change on dryland carbon cycle (Missik et al., 2021; Yao et al., 2020). The motivation of this gap-filling practice was driven by the fact that dryland ecosystems are very sensitive to water availability, functioning as carbon sinks in wet years and carbon sources in dry years (Biederman et al., 2017; Scott et al., 2015), and bias in gap-filled NEE may alter conclusions in sources or sinks of dryland ecosystems in case of relatively long
gaps for eddy covariance data. In addition, different ML algorithms have distinctive internal structures that account for the underlying dependencies of outputs (i.e., NEE) on the inputs (i.e., meteorological variables) in different ways, and uncertainties associated with different ML-based methods can also be assessed (Soloway et al., 2017).

In this study, we evaluate the performance of four commonly used ML algorithms (ANN, KNN, RF, and SVM) in filling the extremely long gaps (i.e., couple months) in the NEE data collected at an EC site over a semi-arid sagebrush ecosystem in the
central Washington, USA, from 2016 to 2019, and assess the uncertainties in the annual NEE introduced by gap-filling methods. In order to fill the extremely long gaps, we propose and examine a two-layer RF-based gap-filling framework (RF-2L) as the RF benefits from better performance and faster run-time than the other ML algorithms (Huang and Hsieh, 2020; Irvin et al., 2021; Kim et al., 2020; Mahabbati et al., 2021).

## 2 Materials and methods

In this section, we first describe the site condition and instruments, as well as the procedures for EC data processing, quality control, and gap identification, following the standard protocol (Mauder and Foken, 2004). We then briefly introduce the four ML algorithms and the proposed framework of RF-2L, as well as the input meteorological variables. Following the previous studies (Moffat et al., 2007; Kim et al., 2020), four different lengths of artificial gaps are generated and used to evaluate the performance of these four ML algorithms, whereas the performance of RF-2L is evaluated with gaps of two months. We also
examine the performance of the algorithms for different time of day scenarios: 1) all the data, 2) daytime and 3) nighttime data. Finally, uncertainties in monthly and annual NEE are quantified by comparing with the results from the MDS method and the ensemble mean of predictions of the ML algorithms.

### 2.1 Site description

The eddy covariance flux tower is located in the Hanford Area in the U.S. State of Washington (AmeriFlux site: US-Hn1;
46°24′32″ N, 119°16′30″ W), and it started to collect data in December 2015 (Gao et al., 2019, 2020a, 2020b; Missik et al., 2019). This semi-arid site is predominantly covered by scattered shrubs and short grasses. Shrub species include *Artemisia tridentata* and *Chrysothamnus viscidiflorus*, and grasses include invasive weedy species (i.e., *Bromus tectorum and Salsola*

*kali*) and native grasses (i.e., *Poa secunda, Pseudoroegneria spicata, and Stipa comate*) (Missik et al., 2019). The long-term (1986-2015) mean annual precipitation was 197 mm (varied between 100 and 300 mm for dry and wet years), most of which occurred late in the fall and early in winter (Missik et al., 2019). The soil texture in the top layer of 30 cm is loamy sand with small rocks and gravel interspersed (Gao et al., 2017; Missik et al., 2019). In this study, the 4-year data from 2016 to 2019 are analyzed. Annual precipitation in these four years were 217, 242, 169, and 210 mm, respectively; and mean annual air temperature were 12.9, 11.4, 12.7, and 11.2°C, respectively (Missik et al., 2021).

**2.2 Eddy covariance and meteorological measurements**

The EC system included a three-dimensional sonic anemometer (CSAT3, Campbell Scientific, Inc.) and an open path gas analyzer (LI-7500A, LI-COR, Inc.), and the EC data were sampled at a rate of 10 Hz. In addition, a variety of microclimate data were measured, including four-component radiation, air temperature and relative humidity, wind speed and direction, precipitation, soil heat flux, and soil temperature and volumetric water content (Gao et al., 2017, 2019, 2020a; Missik et al., 2019). These data were sampled at a rate of 1 Hz and stored as 30-min averages. Further, 15-min meteorological data from two weather stations close to the tower site were obtained from the Washington State University AgWeatherNet (AWN; https://weather.wsu.edu/). The two AWN stations are located within 8 km from the tower. The 15-min data were averaged to half-hourly values to fill gaps in the tower meteorological data of solar radiation ($R_g$), air temperature ($T_{air}$) and relative humidity (RH), vapor pressure deficit (VPD), wind speed (WS), precipitation (P), and soil temperature ($T_{soil}$). Thus, these half-hourly meteorological data for the study period are gap-free.

**2.3 EC data processing, quality control, and gap identification**

Raw 10 Hz EC data were processed using the EddyPro® software (version 7.06, LI-COR Biosciences, USA) to calculate the 30-min average fluxes of $CO_2$ (NEE) and latent (LE) and sensible (H) heat. The data were despiked and filtered for physically impossible values and abnormal diagnostic values of the sonic anemometer and the gas analyzer. The double rotation method was applied to the sonic anemometer data. Block averaging was used to determine the turbulent fluctuations for each 30-min intervals. The fluxes were corrected for the effects of high- and low-pass filtering (Massman, 2000, 2001; Moncrieff et al., 2004) and air density fluctuations (Webb et al., 1980), respectively. The corrected fluxes were quality checked according to Mauder and Foken (2004). After quality checking, the "REddyProc" R package (Wutzler et al., 2018) was used to determine the friction velocity ($u_*$) threshold and NEE data with low turbulence conditions were removed from the dataset.

For simplicity, we assigned a gap score of 2 to gaps due to field operations (e.g., instrument maintenance), electrical and/or instrument failures, a gap score of 1 to gaps due to low data quality (i.e., quality control and $u_*$ filtering), and a score of 0 to flux data with good quality. Only the data with the score of 0 were used to train/test the gap-filling algorithms. Note that, for the data gaps with a score of 1, meteorological data from the flux tower were still available for NEE gap-filling; whereas for

the data gaps with a score of 2, meteorological data from the flux tower also had gaps, and the data obtained from the two nearby AWN stations are thus used for gap filling.

## 2.4 Machine learning algorithms

Four ML algorithms including the ANN, KNN, RF, and SVM were employed and evaluated for filling NEE gaps. In the following sections, we briefly describe the characteristics and implementation of each ML algorithm. The required parameters in each algorithm (e.g., the number of nodes in each hidden layer for the ANN; k value for the KNN) are optimized using the "caret" R package (Kuhn et al., 2020) with a 10-fold cross-validation repeated ten times.

### 2.4.1 Artificial neural network (ANN)

The ANN algorithm has been successfully applied for filling NEE gaps in various ecosystems (Baldocchi & Sturtevant, 2015; Knox et al., 2016; Moffat et al., 2007; Papale & Valentini, 2003; Tramontana et al., 2016). In this study, we employed the "neuralnet" R package (Günther and Fritsch, 2010), the resilient backpropagation algorithm that has proven to be capable of gap-filling flux data (Dengel et al., 2013; Jammet et al., 2015; Kim et al., 2020; Knox et al., 2016). The required parameters for the ANN algorithm include the number of hidden layers and the number of nodes in each layer. Here, based on the parameter tuning results, we use two hidden layers with 12 and 10 nodes in the first and second hidden layers. We train the neural network 1000 times, and the mean prediction results of the top 20 runs based on their training and testing $R^2$ values are used to fill NEE gaps (Baldocchi & Sturtevant, 2015; Knox et al., 2016).

### 2.4.2 K-nearest neighbours (KNN)

The KNN algorithm (Fix and Hodges, 1951) is a non-parametric ML approach and has been used in many applications. For example, Chen et al. (2012) applied the KNN algorithm for filling latent heat flux gaps, which fills the data gaps based on a certain attribution of $k$ neighbours in the feature space. In this study, we use the "caret" R package (Kuhn et al., 2020) to build the KNN where a suitable $k$ value needs to be determined. Here, the optimized k value is 9.

### 2.4.3 Random forest (RF)

The RF algorithm (Breiman, 2001) has been applied for upscaling flux data to regional (Xu et al., 2018) and global (Jung et al., 2017; Zeng et al., 2020) scales and recently for gap-filling flux data (Huang and Hsieh, 2020; Kim et al., 2020). The RF algorithm uses bootstrap aggregation and feature randomness when generating each individual tree to try to create many independent decision trees that operate as an ensemble of the prediction results. In this study, we create 500 regression trees for each case using the "randomForest" R package (Liaw and Wiener, 2002), in which the tuning parameter is the number of randomly selected predictors (i.e., mtry equals 7). In addition, the RF allows the estimation of relative importance of input variables, and such a feature has been utilized in previous studies to help interpret the results (Irvin et al. 2021; Kim et al., 2020).

### 2.4.4 Support vector machine (SVM)

The SVM algorithm (Cortes and Vapnik, 1995) has also been applied for gap-filling (Huang and Hsieh, 2020; Kim et al., 2020) and upscaling (Xu et al., 2018) flux data. The SVM algorithm can convert nonlinear regressions into linear regressions by projecting the original finite-dimensional space into the much higher-dimensional space with a predefined kernel function. In this study, we use the radial basic kernel function and the "kernlab" R package (Karatzoglou et al., 2004) where the tuning parameters include the inverse kernel width (i.e., sigma = 0.13) and the cost regularization parameter (i.e., C = 27).

### 2.4.5 A two-layer RF based gap-filling framework (RF-2L) for extremely long gaps

Numerous studies have suggested that the performance of ML algorithms decreases with increasing the gap length, and that the ML algorithms is only reliable for gaps shorter than couple weeks (Huang and Hsieh, 2020; Irvin et al., 2021; Kim et al., 2020; Mahabbati et al., 2021). In order to fill the extremely long gaps (i.e., couple months), we propose and examine a two-layer RF-based gap-filling framework (RF-2L) because the RF outperforms most of the other ML algorithms in gap-filling the half-hourly fluxes over various ecosystems (Huang and Hsieh, 2020; Irvin et al., 2021; Kim et al., 2020; Mahabbati et al., 2021) and is more time efficient than the ANN (Irvin et al., 2021). The procedures of the RF-2L include: 1) train the RF model using the half-hourly data and fill the NEE gaps shorter than 7 days; 2) calculate daily means of the input variables and the partially filled NEE data; and 3) train the RF model using the daily data and fill the gaps in the daily NEE data.

### 2.5 Input variables

Besides the above mentioned meteorological variables, the input variables for the ML algorithms also include the normalized difference vegetation index (NDVI) and enhanced vegetation index (EVI) from the Moderate Resolution Imaging Spectroradiometer (MODIS), and three fuzzy variables (i.e., decimal day of year, and sine and cosine functions to represent seasonal changes) following Moffat et al. (2007). We obtained the NDVI and EVI data around the flux tower location from the MOD13Q1 version 6 data product (https://lpdaac.usgs.gov/products/mod13q1v006/) at 16-day temporal and 250-m spatial resolutions (Didan, 2015). The 16-day NDVI and EVI data were resampled to 30 minutes using cubic spline interpolation. Note that soil water content (SWC) and groundwater table are not included in the inputs because 1) the SWC measurements at these AWN stations had some issues for the study period and could not be used to fill the long gaps in tower SWC data, and 2) the station is primarily fed by rainfall without groundwater access (Missik et al., 2019, 2021).

With the RF, we estimate the relative importance of the input variables for the three different time of day scenarios, respectively (Fig. 1). For the model trained using all the data and the daytime data, $R_g$ is the most important variable, while for the model trained using the nighttime data, $T_{air}$ and RH are the most important variables. Overall, the estimated variable importance indicates that, for the three different time of day scenarios, the meteorological inputs except $R_g$ for the nighttime data play a comparable role in the trained gap-filling models. Note that there exists collinearity among the various variables, and thus the

model performance might approach a plateau with certain input variables and increase slightly with increasing the number of inputs, resulting in a slower, less efficient model. However, to be consistent with previous studies (e.g., Kim et al., 2020; Moffat et al., 2007), we also include all available meteorological variables as the inputs to train and evaluate the models.

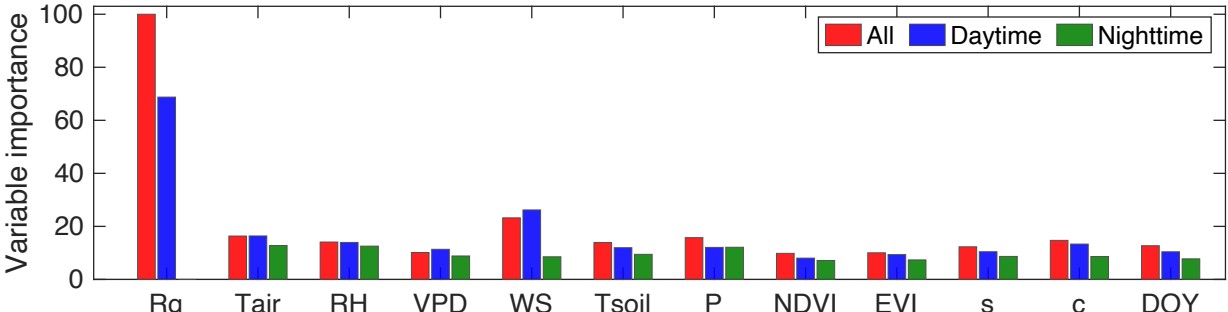

**Figure 1** Variable importance for the random forest trained without separating the daytime and nighttime data (All), and separately for daytime and nighttime data. The variables DOY, c and s indicate decimal day of year, cosine and sine functions, respectively.

### 2.6 Artificial gap scenarios and performance evaluation

In order to evaluate the gap-filling algorithms, artificial gaps with different lengths are randomly generated in the original flux data, accounting for approximately 10~12% of the NEE data with a score of 0. Following Moffat et al. (2007) and Kim et al. (2020), we consider four artificial gap lengths: one hour (1-H), one day (1-D), one week (1-W), and two months (2-M). The variations in the number of data points artificially removed are because after quality control, it is hard to locate even couple days of data without any missing points (Section 3.1). In order to reduce the potential effect of uneven sample sizes and gap positions on the performance evaluation, each gap length scenario is permuted 10 times, resulting in 40 distinct time series with artificial gap scenarios.

For each artificial gap scenario, we traine the models separately with only daytime or nighttime data, and with both daytime and nighttime data. The performance of gap-filling algorithms is evaluated by comparing the estimated values ($e_i$) with the measured values ($m_i$) for the artificial gaps. Four commonly used performance metrics are used including the coefficient of determination ($R^2$), the root mean square error (RMSE), mean absolute error (MAE), and the bias error (BE):

$$R^2 = \frac{\left(\sum_{i=1}^{n}(e_i-\bar{e})(m_i-\bar{m})\right)^2}{\sum_{i=1}^{n}(e_i-\bar{e})^2 \sum_{i=1}^{n}(m_i-\bar{m})^2} \tag{1}$$

$$\text{RMSE} = \sqrt{\frac{\sum_{i=1}^{n}(e_i-m_i)^2}{\sum_{i=1}^{n}(m_i)^2}} \tag{2}$$

$$\text{MAE} = \frac{1}{n}\sum_{i=1}^{n}|e_i - m_i| \tag{3}$$

$$BE = \frac{1}{n}\sum_{i=1}^{n}(e_i - m_i) \hspace{6cm} (4)$$

where the overbar denotes the mean value.

Besides the above statistic metrics, we also calculate the probability density functions of the measured and estimated values to examine the performance of ML algorithms in terms of different ranges of NEE values. Here, the probability density functions are calculated as the binned density distribution of measured (estimated) NEE values divided by the bin width (0.2 µmol m-2 s-1).

Previous studies also suggested that the model errors of gap-filling algorithms should approach the measurement random errors of the EC method (Kim et al., 2020; Moffat et al., 2007; Richardson et al., 2008). In this study, we compare the model errors with the random measurement errors as a reference. Here, the random measurement error is estimated following the method proposed by Finkelstein and Sims (2001).

## 2.7 Uncertainty Estimation

With the 40 distinct artificial gap scenarios, we obtain 40 gap-filled NEE time series for each method. By replacing the artificial gaps with the observed data, these time series datasets allow for an evaluation of the model self-agreement and reliability in filling the actual gaps. Monthly and annual NEE are then computed from the gap-filled flux time series. The model self-agreement can be evaluated by investigating the mean standard deviations of the monthly NEE (Soloway et al., 2017), whereas the uncertainties in the monthly and annual NEE can be assessed by comparing the monthly and annual NEE obtained by each ML algorithm with their ensemble means (Irvin et al., 2021; Kim et al., 2020; Soloway et al., 2017). Here, the flux time series gap-filled separately for the daytime and nighttime periods are combined to determine the monthly and annual NEE.

## 3 Results

### 3.1 NEE data gap evaluation

At the US-Hn1 site, different lengths of gaps were found in NEE data during the four years from 2016 to 2019 (Fig. 2). Gaps with short to medium lengths were usually caused by low data quality (i.e., gap score of 1), whereas gaps with medium to extremely long lengths were mostly due to electrical and/or instruments failures (i.e., gap score of 2). Data gaps with a score of 1 frequently occurred in nighttime because $u_*$ filtering mainly removes nighttime data. On average, gaps with scores of 1 and 2 accounted for about 28.4% and 30.2% of half-hour NEE data, respectively (Table 1). There were more extremely long gaps in 2018 and 2019 than 2016 and 2017 due to power failures in winter and early spring. Therefore, it is worth to examining the performances of gap-filling algorithms in filling different lengths of gaps. Overall, about 43.7% of NEE data were available to calibrate and validate the gap-filling methods.

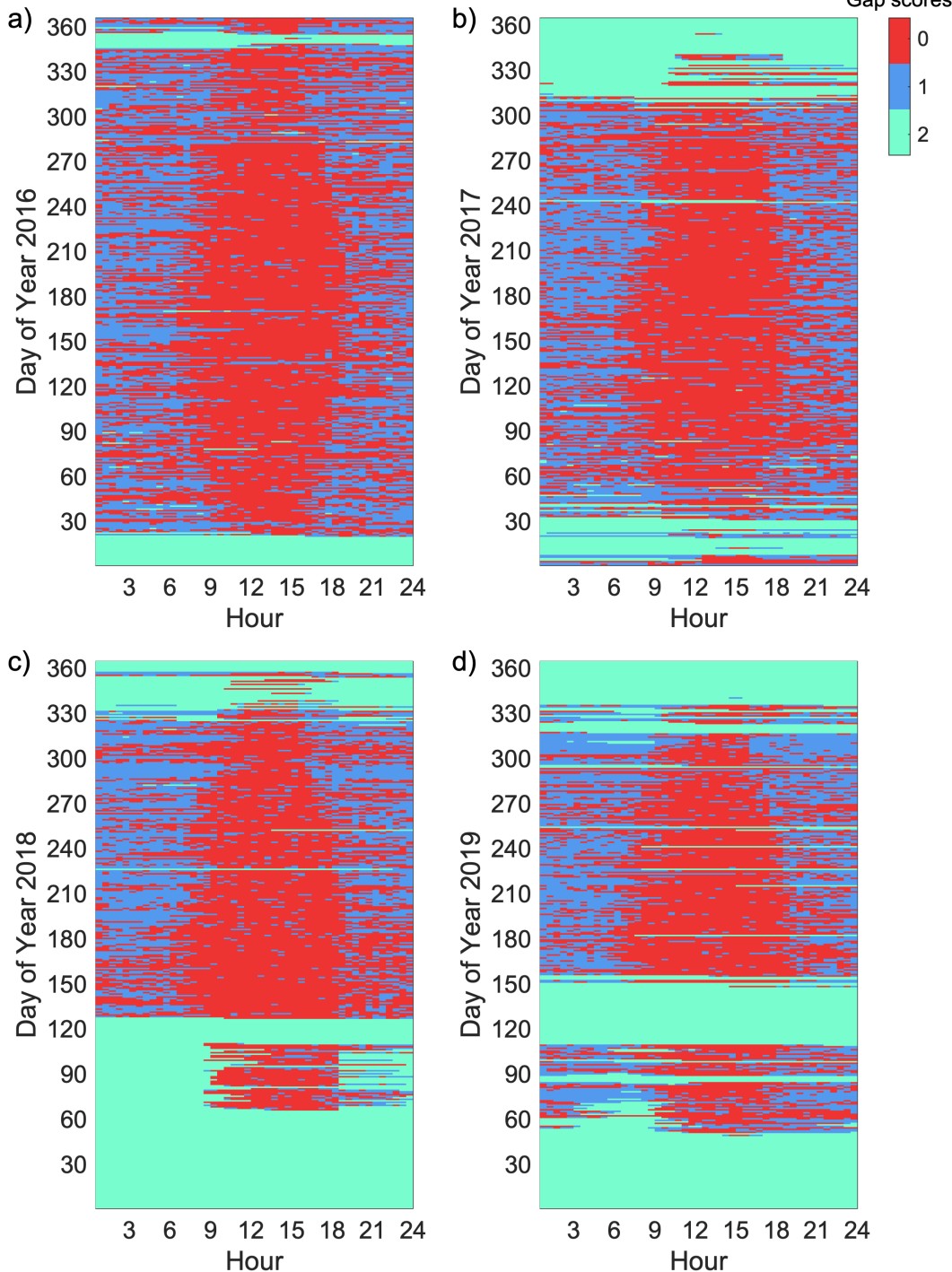

**Figure 2** Distribution of NEE data gaps by day and hour from 2016 to 2019. NEE data gaps were classified as non-gaps (gap score: 0), gaps due to low data quality (gap score: 1), and gaps due to electrical and/or instrument failures (gap score: 2).

**Table 1** Percentage of NEE data with different gap scores (0/1/2) for daytime and nighttime from 2016 to 2019. NEE gaps are classified as non-gaps (gap score: 0), gaps due to low data quality (gap score: 1), and gaps due to electrical and/or instrument failures (gap score: 2).

|  | Daytime | Nighttime | All |
|---|---|---|---|
| 2016 | 36.1% / 9.6% / 2.7% | 19.9% / 26.1% / 5.6% | 56.0% / 35.7% / 8.3% |
| 2017 | 31.5% / 9.8% / 7.1% | 15.4% / 22.3% / 13.9% | 46.9% / 32.1% / 21.0% |
| 2018 | 27.8% / 6.8% / 14.1% | 10.9% / 16.5% / 23.9% | 38.7% / 23.3% / 38.0% |
| 2019 | 23.4% / 7.1% / 17.9% | 10.0% / 18.1% / 23.5% | 33.4 % / 25.2 % / 41.4% |
| Overall | 29.7% / 8.4% / 10.5% | 14.0% / 20.7% / 16.7% | 43.7% / 29.1% / 27.2% |

### 3.2 Performance of ML gap-filling algorithms with different gap lengths

We first train and evaluate the ML algorithms without separating the daytime and nighttime data (i.e., using all the data). The overall performance of each algorithm degrades as the gap length increases, while the RF slightly outperforms the other algorithms for all the gap scenarios (Fig. 3 and Table 2). For the gap length of one hour (1-H), all the four ML algorithms have the highest $R^2$ and the lowest RMSE, and MAE. As the gap length increases, $R^2$ for all ML algorithms decreases and RMSE (MAE) increases. For 1-H, the RF has $R^2$ and RMSE of 0.77±0.02 and 0.47±0.02, respectively, whereas for the gap length of

two months (2-M), the $R^2$ decreases to 0.60±0.22, and RMSE increase to 0.62±0.17, respectively. The magnitudes of BE for all ML algorithms also increase with the increasing gap lengths. The performance of these ML algorithms in NEE gap-filling at this semi-arid sagebrush site is comparable to that at some grassland sites (Huang and Hsieh, 2020), but lower than that at forest and cropland sites (Huang and Hsieh, 2020; Moffat et al., 2007). The relatively low performance might be caused by the spatial complexities of the targeted $CO_2$ flux and input meteorological drivers within the flux footprints (Chu et al., 2021;

Stoy et al., 2013), especially for the scattered sagebrush ecosystems.

Following Moffat et al. (2007), we also perform the algorithm training and evaluation separately for daytime and nighttime data. For the daytime data, the performance of each algorithm is similar to that for all the data, whereas the algorithm performance for the nighttime data is degraded with $R^2$ of 0.1–0.2 and RMSE of 0.7–0.8, similar to the results for some forest sites in Moffat et al. (2007). The poor performance of the gap-filling algorithms for the nighttime data is primarily attributed

to the shortage of available nighttime data for training the models (Fig. 2 and Table 1). However, the change of BE with the increased gap length at night is relatively small compared to that for the daytime data, especially for the ANN and RF. In addition, BE for long gaps (e.g., 2-M) has opposite signs for the daytime and nighttime data, resulting in a smaller BE for all the data.

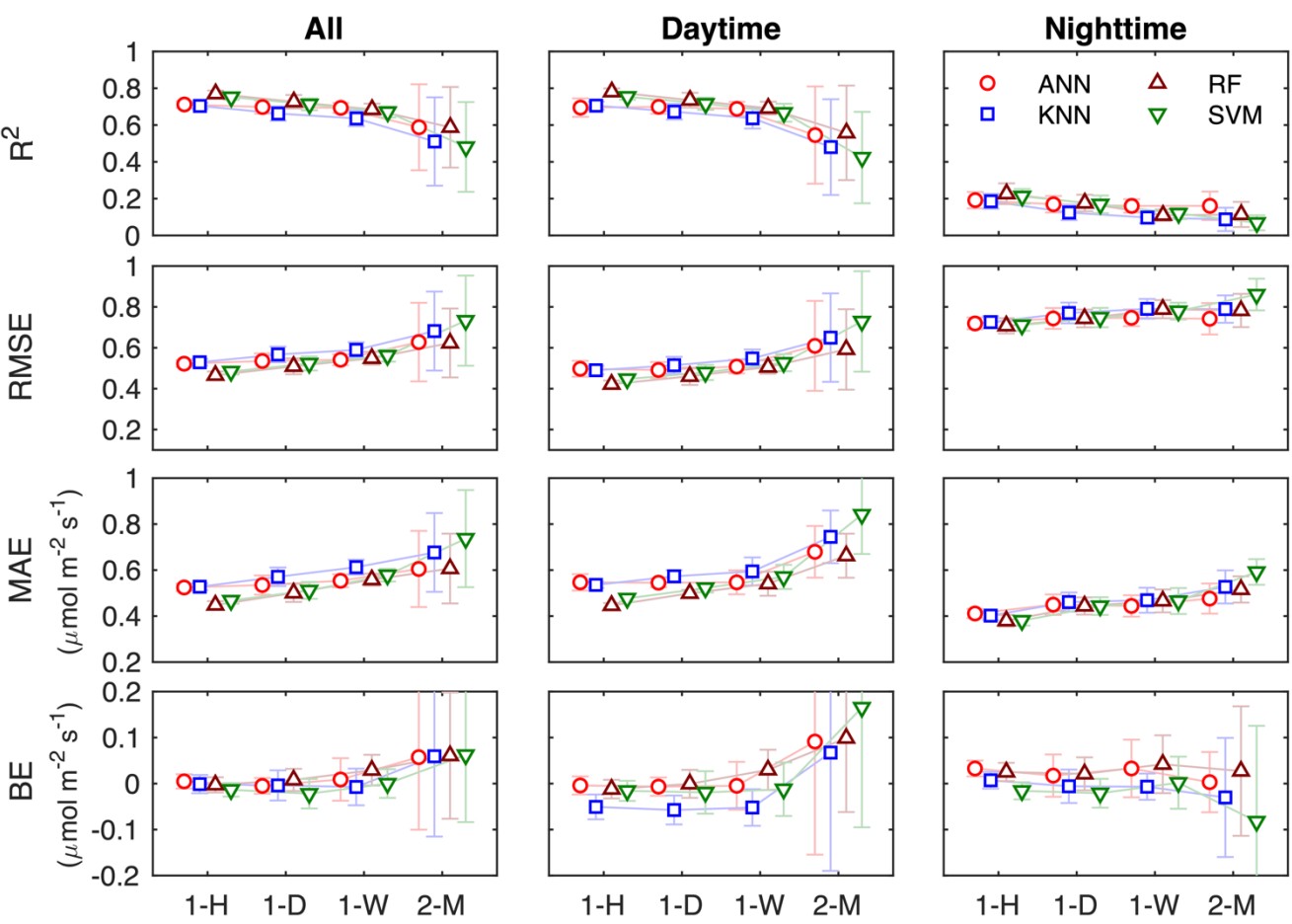

**Figure 3** Performance of NEE gap-filling algorithms for the four gap length scenarios (i.e., one-hour (1-H), one-day (1-D), one-week (1-W), and two-month (2-M), respectively).


**Table 2** Performance metrics of the four ML gap-filling algorithms for the four gap length scenarios. The ML algorithms are trained and evaluated without separating daytime and nighttime data.

| | Gap length scenarios | $R^2$ | RMSE | MAE ($\mu mol\ m^{-2}\ s^{-1}$) | BE ($\mu mol\ m^{-2}\ s^{-1}$) |
|---|---|---|---|---|---|
| ANN | One-hour | 0.71±0.02 | 0.52±0.02 | 0.50±0.01 | -0.005±0.016 |
| | One-day | 0.70±0.02 | 0.54±0.04 | 0.51±0.02 | 0.005±0.017 |
| | One-week | 0.69±0.03 | 0.54±0.03 | 0.51±0.04 | -0.009±0.046 |
| | Two-month | 0.59±0.23 | 0.63±0.19 | 0.62±0.07 | -0.057±0.158 |
| KNN | One-hour | 0.71±0.02 | 0.53±0.02 | 0.50±0.01 | 0.001±0.020 |
| | One-day | 0.66±0.04 | 0.57±0.04 | 0.55±0.02 | 0.004±0.033 |
| | One-week | 0.63±0.05 | 0.59±0.04 | 0.57±0.06 | 0.007±0.040 |
| | Two-month | 0.51±0.24 | 0.68±0.19 | 0.69±0.08 | -0.059±0.175 |
| RF | One-hour | 0.77±0.02 | 0.47±0.02 | 0.43±0.01 | 0.003±0.017 |
| | One-day | 0.73±0.04 | 0.51±0.04 | 0.48±0.02 | -0.008±0.024 |
| | One-week | 0.68±0.03 | 0.55±0.03 | 0.52±0.05 | -0.029±0.033 |
| | Two-month | 0.60±0.22 | 0.62±0.17 | 0.61±0.05 | -0.061±0.137 |
| SVM | One-hour | 0.75±0.02 | 0.48±0.02 | 0.45±0.01 | 0.014±0.015 |
| | One-day | 0.71±0.03 | 0.52±0.03 | 0.49±0.02 | 0.022±0.032 |
| | One-week | 0.67±0.03 | 0.56±0.03 | 0.53±0.04 | -0.000±0.031 |
| | Two-month | 0.47±0.25 | 0.73±0.22 | 0.77±0.16 | -0.062±0.146 |


### 3.3 Comparison of probability density functions (PDFs) between measured and estimated NEE

Figure 4 shows the comparison of the probability density functions (PDFs) between the measured and estimated NEE of the gap-filling algorithms. The estimated NEE by the gap-filling algorithms has similar shape of PDFs, which is also quite similar to that of the measured NEE with some difference in amplitudes. At this site, the measured half-hourly NEE ranges from -6 to 4 $\mu mol\ m^{-2}\ s^{-1}$, while the estimated NEE varies from approximately -5 to 2 $\mu mol\ m^{-2}\ s^{-1}$. For all the data, the PDFs of the estimated NEE present two peaks at around -0.2 and 0.2 $\mu mol\ m^{-2}\ s^{-1}$, whereas the PDF of the measured NEE only has one peak at around 0.2 $\mu mol\ m^{-2}\ s^{-1}$. That means that the PDFs of the estimated NEE have higher amplitude than the measured NEE in the range of -0.6 to 0.0 $\mu mol\ m^{-2}\ s^{-1}$. However, in the range of -2.0 to -0.8 $\mu mol\ m^{-2}\ s^{-1}$, the amplitude of the PDFs of the estimated NEE is lower than that of the measured NEE. The PDFs of the estimated NEE for the daytime data show one peak at around -0.2 $\mu mol\ m^{-2}\ s^{-1}$ with their shapes similar to those for all the data in the range with the negative NEE values; for the nighttime data, the estimated NEE has a narrower shape of PDF than that of the measured NEE. These results suggest



that the gap-filling algorithms underestimate the magnitudes of NEE in the range of -2.0 to 0.0 µmol m$^{-2}$ s$^{-1}$ and the magnitudes of the peak values of NEE.

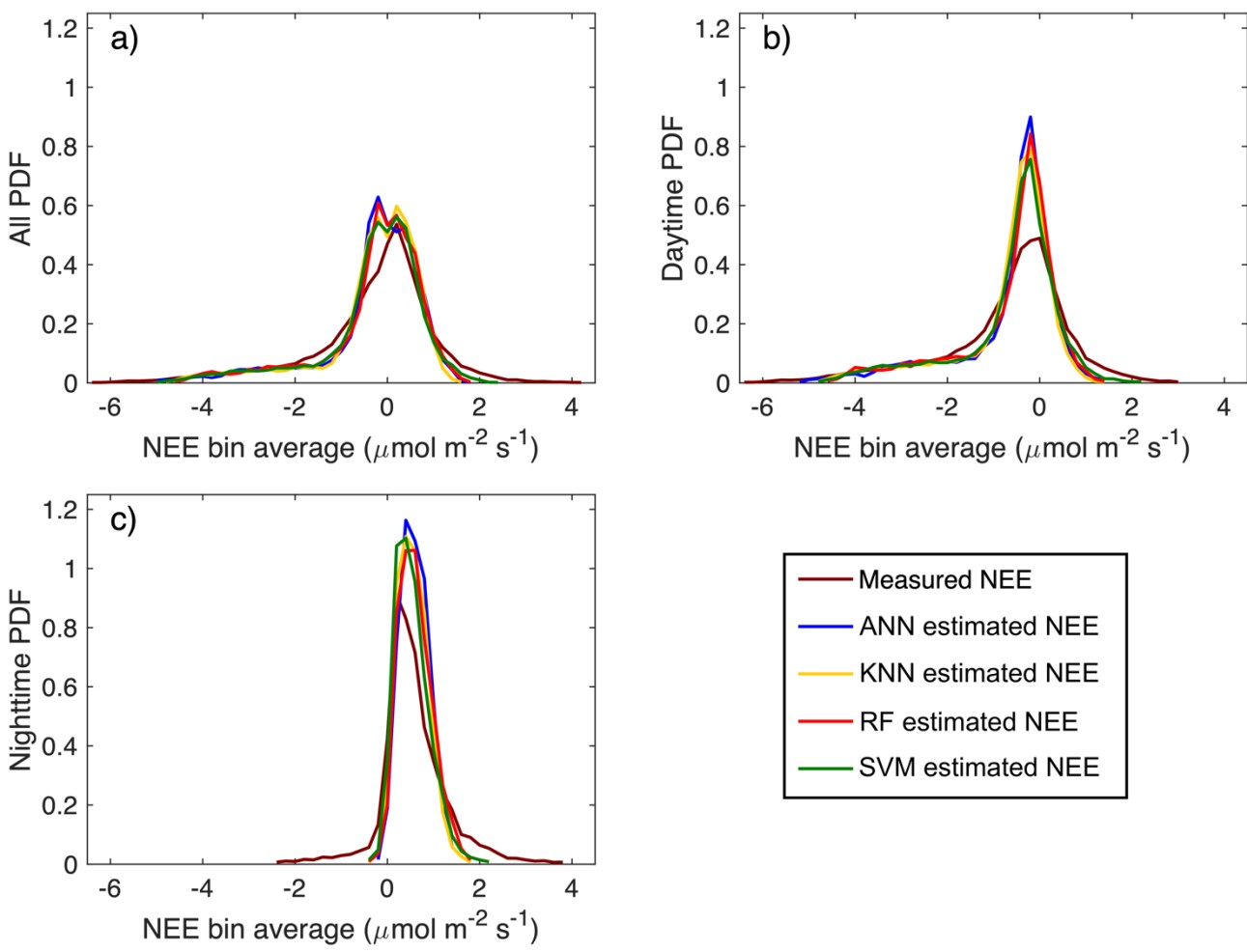

 **Figure 4** Comparison of probability density functions (PDFs) for the measured and estimated NEE of the gap-filling algorithms using daytime and nighttime data together and separately. The PDFs are estimated from the binned density distributions divided by the bin width. The bin width is 0.2 µmol m$^{-2}$ s$^{-1}$ and bins with less than 10 data points are excluded.

### 3.4 Comparison of measurement error and absolute error of ML algorithms

Figure 5 shows the absolute errors as a function of bin-averaged NEE values for the gap-filling algorithms with the estimated
 random measurement errors (Finkelstein and Sims, 2001) as a reference. For all the data, the binned absolute errors for the ML algorithms are quite close to each other, and they are also close to the random errors in the range of negative NEE values but slightly deviated from the random errors in the range of positive NEE values. Note that the large deviations from the

random errors at the edges are most likely due to the small number of data points as illustrated in Fig. 4. The binned absolute errors for the daytime data are close to the random errors in the NEE range of about -4.0 to 1.0 µmol m$^{-2}$ s$^{-1}$; whereas the binned absolute errors for the nighttime data are consistently higher than the random errors. For all the data, the mean value of the random errors is 0.56 µmol m$^{-2}$ s$^{-1}$, while the MAE of the ML algorithms is 0.55, 0.59, 0.53, and 0.58 µmol m$^{-2}$ s$^{-1}$ for the ANN, KNN, RF, and SVM, respectively. For the daytime data, the MAE of the ML algorithms (0.59, 0.62, 0.55, and 0.62 µmol m$^{-2}$ s$^{-1}$) is all smaller than the mean value of the random error (0.66 µmol m$^{-2}$ s$^{-1}$); whereas for the nighttime data, the MAE of the ML algorithms (0.46, 0.48, 0.47, and 0.49 µmol m$^{-2}$ s$^{-1}$) is all larger than the mean value of the random error (0.32 µmol m$^{-2}$ s$^{-1}$). Overall, the RF and ANN have better performance in filling NEE gaps at this semi-arid site, especially for the daytime data, although the RF is more time efficient than the ANN. In addition, all the four ML algorithms have low performance in gap-filling the nighttime data, though the BE for the nighttime data is relatively small compared to the daytime data mostly due to the low magnitudes of the nighttime NEE.

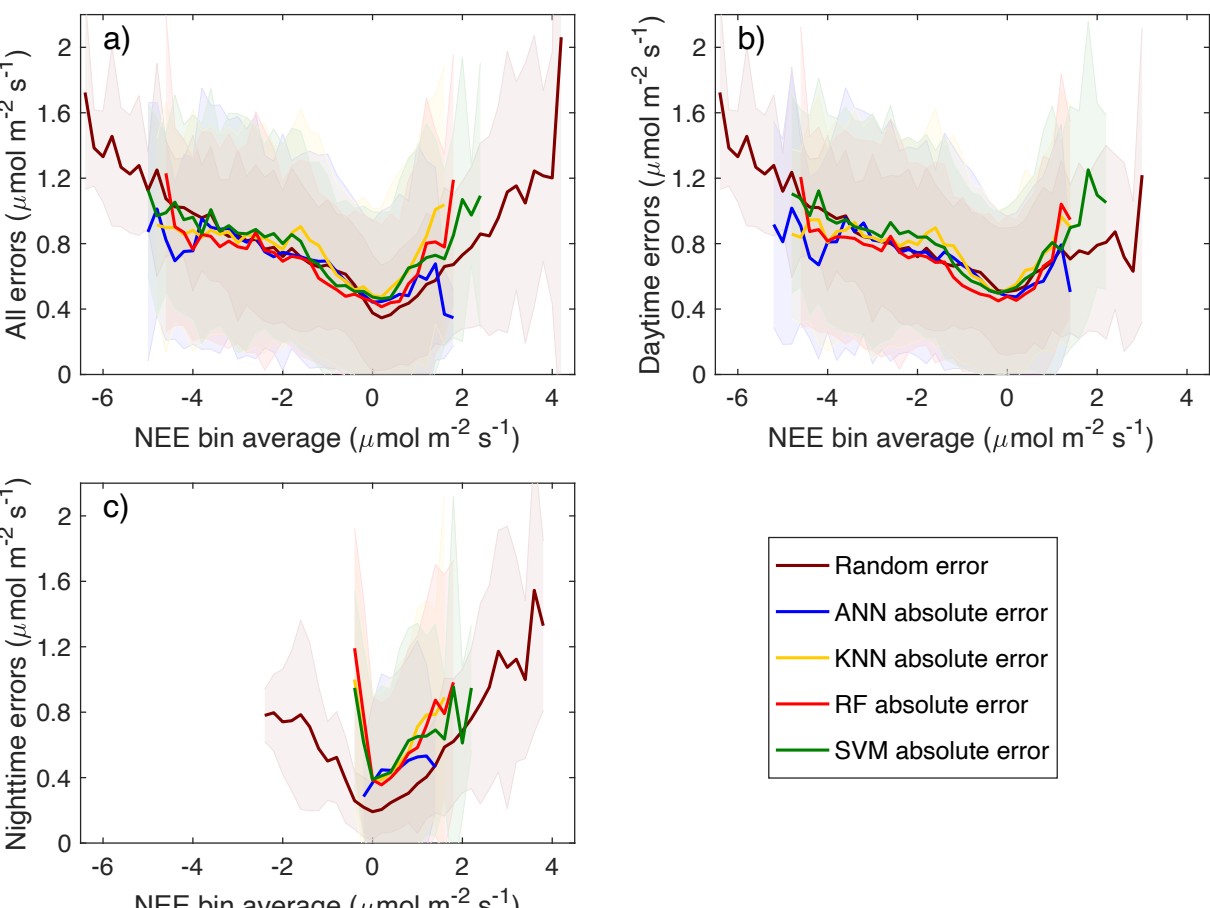

 **Figure 5** Comparison of NEE measurement and model uncertainties of the gap-filling algorithms using the data and the daytime and nighttime data. The random measurement error is estimated by Finkelstein and Sims (2001). The bin width is 0.2 $\mu$mol m$^{-2}$ s$^{-1}$ and bins with less than 10 data points are excluded.

## 4 Discussion

### 4.1 Uncertainties in carbon budgets caused by gap-filling

We now examine uncertainties in carbon budgets caused by gap filling with different training dataset and different methods. Figure 6 compares the monthly NEE of the gap-filled data during 2016-2019. The subscript A denotes that the daytime and nighttime NEE data are gap-filled together using the ML algorithms trained with all the data, and DN denotes that the daytime and nighttime data are gap-filled separately using the trained ML algorithms and then combined together to determine the monthly NEE. The error bar denotes one standard deviation of the monthly NEE of the 40 gap-filled time series. Here the gap-filled NEE using the most-commonly used MDS method is also included as a reference for comparison.

With the 40 gap-filled NEE time series for each method, we first investigate the self-agreement of each method as the method with good self-agreement should have small variations in the accumulative NEE among different trails. For months with gaps less than 7 days (Fig. 2; e.g., February to October in 2016), all the four ML algorithms have good self-agreement with the mean standard deviations of the monthly NEE ranging from 0.4 (ANN) to 1.0 (SVM) g C m$^{-2}$. Both the RF and KNN have mean standard deviations of approximately 0.7 g C m$^{-2}$ during these months. For months with long gaps (i.e., > 7 days), the mean standard deviations of the monthly NEE are 1.2, 1.3, 1.5, and 3.3 g C m$^{-2}$ for the ANN, KNN, RF, and SVM, respectively. From this perspective, the ANN is the most reliable method in gap-filling because the predicted values are averages of the best 20 runs (Section 2.4.1). In other words, most of the ML algorithms are quite consistent in filling the gaps, and the differences in the monthly NEE caused by changes in training dataset are less than 1.5 g C m$^{-2}$ except for the SVM.

The uncertainties in the monthly NEE as a result of the differences in the methods are now assessed with the monthly NEE from the MDS as a reference. The difference among the methods ranges from 0.2 to 1.3 g C m$^{-2}$ for months with gaps less than 7 days, and changes from 0.8 to 10.2 g C m$^{-2}$ for months with long gaps, which is 0.8 to 4.8 g C m$^{-2}$ without including the SVM. For the ANN and RF, the differences in the monthly NEE between A and DN are usually quite small for all the months, whereas the KNN presents opposite signs for A and DN in months with long gaps, which means that the KNN is unable to handle the daytime and nighttime data together. Overall, for months with short to medium gaps (i.e., < 7 days), there is no significant difference in the monthly NEE among the methods including the MDS method. For months with long gaps (i.e., > 7 days), the MDS method usually fails, and the ANN and RF have the best performance, and they have the potential to handle the daytime and nighttime data gap-filling together, as also supported by the distribution of estimated variable importance (Fig. 1).

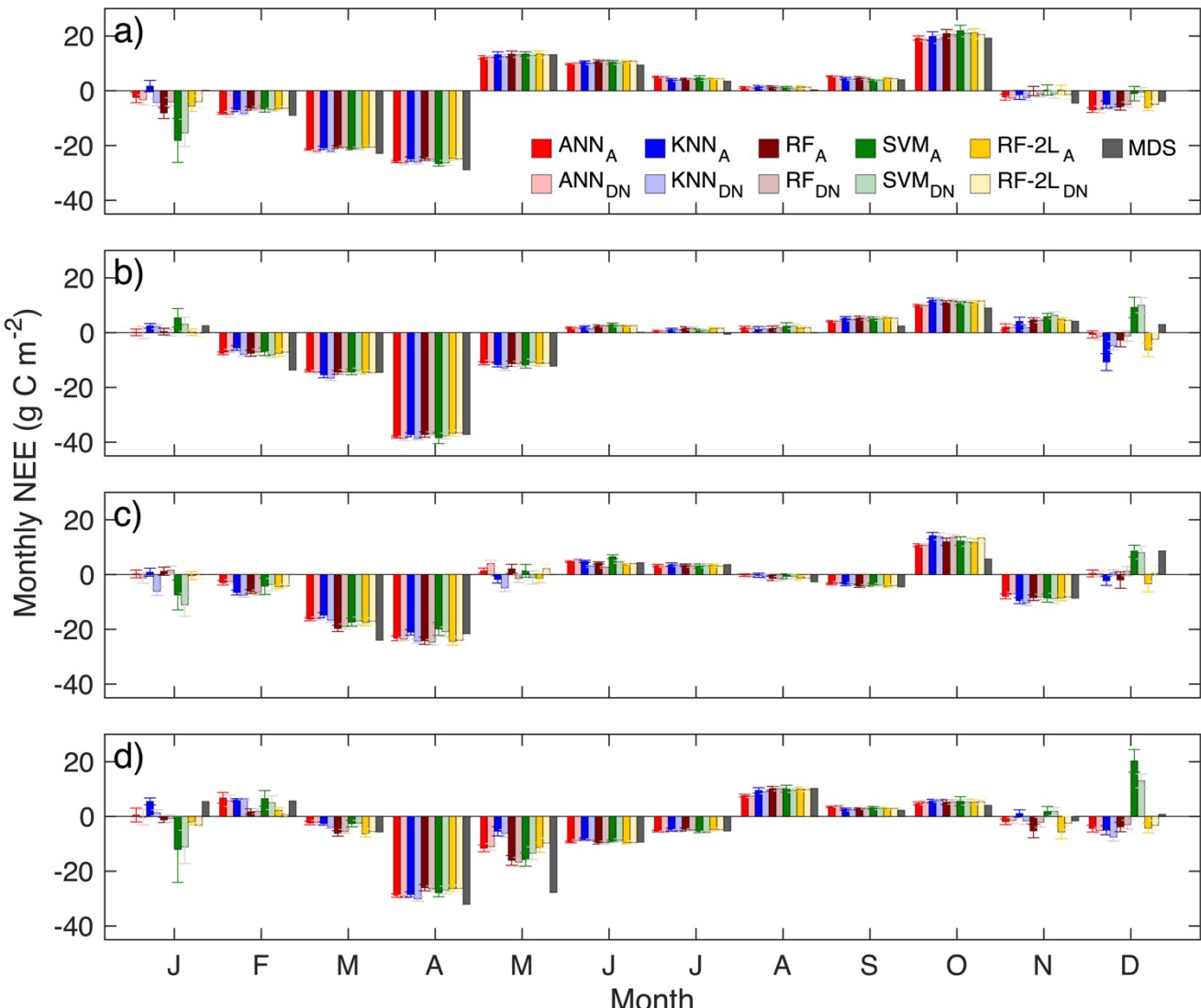

**Figure 6** Monthly total net ecosystem exchange (NEE, g C m$^{-2}$) at the US-Hn1 in (a-d) 2016-2019. The subscript A denotes that the daytime and nighttime NEE data are gap-filled together using the ML algorithms trained with all the data, and DN denotes that the daytime and nighttime data are gap-filled separately and then combined to determine the monthly total values. RF-2L denotes the proposed two-layer RF based gap-filling framework, and MDS is the marginal distribution sampling algorithm. The error bar denotes one standard deviation of the monthly total NEE of the 40 gap-filled time series.

The uncertainties in the annual total NEE are estimated by comparing the annual NEE obtained by each ML algorithm with their ensemble means (Fig. 7). Obviously, both the KNN and SVM are largely apart from the ensemble means, whereas the ANN and RF are relatively close to the ensemble means except for the RF in 2019. The mean standard deviations of the 40 trials for the annual NEE by ANN range from 2.5 to 4.3 g C m$^{-2}$, the mean standard deviations by the RF vary from 3.4 to 7.1

g C m$^{-2}$, and the mean standard deviations by the KNN and SVM vary from 5.1 to 13.9 g C m$^{-2}$. The differences between the ANN and RF are within ±8.4 g C m$^{-2}$; and the differences between A and DN are less than 1.7 and 4.5 g C m$^{-2}$ for the ANN and RF, respectively. The overall uncertainties in the annual NEE caused by the ANN and RF are usually less than 15.5 g C m$^{-2}$, while the uncertainties can be as large as 27.2 g C m$^{-2}$ if including the KNN and SVM. Therefore, it is recommended to use the ensemble mean of the ANN and RF as the best estimate of the annual NEE at the semi-arid sagebrush site. The annual mean NEE by the ANN and RF is -15.4±4.7, -50.0±9.1, -31.4±7.2, and -40.3±8.7 g C m$^{-2}$ for 2016, 2017, 2018, and 2019, respectively. In addition, the annual total NEE by the MDS is about 5.6~15.6 g C m$^{-2}$ larger than that by the ANN and RF.

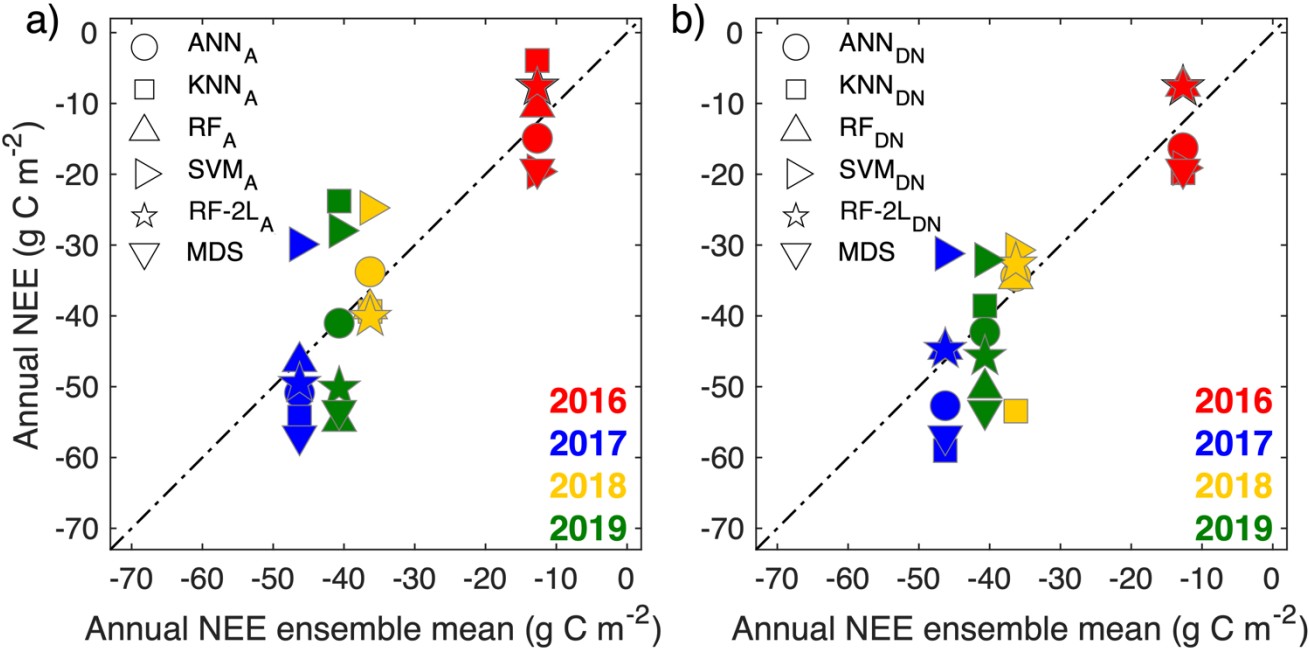

**Figure 7** Comparison of the annual mean NEE from the ML algorithms and their ensemble mean. The subscript A denotes that daytime and nighttime NEE data are gap-filled together using the ML algorithms trained with all the data, and DN denotes that the daytime and nighttime data are gap-filled separately and then combined to determine the monthly total values.

## 4.2 Performance of the two-layer RF based gap-filling framework (RF-2L)

Following the same procedures as for evaluating the ML algorithms above, the performance of RF-2L in filling the long gaps (i.e., two months) in the daytime and nighttime daily means are also accessed. Using a 10-fold cross-validation repeated ten times, the R$^2$ of the second layer RF model is 0.78, 0.85, and 0.77 for all the data, the daytime data, and the nighttime data, respectively. The mean BE for this framework is 0.04 g C m$^{-2}$ day$^{-1}$, which is slightly smaller than that of the original RF with the half-hourly NEE data (0.06 g C m$^{-2}$ day$^{-1}$).

The uncertainties of the RF-2L are accessed by comparing the monthly and annual NEE with other ML algorithms. As shown in Figs. 6 and 7, the monthly and annual NEE by the RF-2L are quite close to those by the RF. For A (i.e., trained by all the data), the difference in the annual mean NEE ranges from 0.0 to 2.2 g C m$^{-2}$, whereas for DN (i.e., trained by the separated daytime and nighttime data), the difference varies from 0.8 to 3.3 g C m$^{-2}$. This test suggests that it is not necessary to fill all the gaps in the half-hourly NEE data if the focus is on assessing the uncertainties in annual mean NEE and interannual variability. Therefore, the RF-2L provides an alternative in filling extremely long gaps to characterize annual carbon budgets and interannual variability in dryland ecosystems. In addition, the performance of the different ML algorithms is quite consistent when filling short to medium gaps (e.g., < 7 days), and thus a promising extension of the proposed approach is that using the ensemble mean of multiple methods as the input of the second RF layer, which may have the potential to lower the uncertainties in the gap-filled data. Of course, other reliable algorithms can also be applied in the second layer to reduce bias estimation caused by long gaps.

## 5 Conclusions

The performance of the four ML algorithms in filling the NEE data gaps is evaluated at a semi-arid sagebrush ecosystem site. Due to the relatively small range of NEE variations, the overall performance of these gap-filling algorithms at this site is lower than that at other forest sites, but comparable to that at other grassland sites. The RF algorithm outperforms the other algorithms in terms of the overall performance. It is not necessary to train the model separately for daytime and nighttime data when using the ANN and RF algorithms. The uncertainties in the monthly and annual NEE due to the gap-filling approaches are evaluated by the standard deviations of monthly NEE of multiple trials for each method and also accessed by the difference in the monthly NEE by the methods. With the ANN and RF, the uncertainties in annual NEE are usually within 16 g C m$^{-2}$ at this semi-arid sagebrush site. Extremely long gaps in half-hourly NEE data due to power failures cannot be confidently filled by either of the methods because of the high uncertainties in R$^2$ and RMSE, and thus we propose and test a two-layer RF based gap-filling framework. With this framework, the improvement in model performance is significant, especially for the nighttime data. Therefore, it is recommended that the two-layer RF based framework (RF-2L) should be used if there are extremely long gaps existed in the NEE dataset and if there is a need to investigate its annual and interannual variability. However, it is hard to assess the uncertainties caused by bias in the gap-filled meteorological variables using the current approaches and study design, which need to be explored in future studies.

**Code availability**

The code used in this study are available from J.Y. and J.H. upon request.

**Data availability**

The data used in this study are deposited in a public domain repository at https://doi.org/10.6084/m9.figshare.14747952.

**Author contribution**

J.Y., Z.G., and H.L. contributed equally to this work. J.H designed the study with substantial input from all coauthors. Z.G. and H.L. conducted the field work and obtained and processed the EC data. J.Y. performed the gap-filling and analyzed the
results. J.Y and Z.G. drafted the manuscript. J.H., H.L., and G.W. contributed to the result analysis and interpretation. All authors commented on and approved the final manuscript.

**Competing interests**

The authors declare that they have no conflict of interest.

**Acknowledgments**

We acknowledge support by the Second Tibetan Plateau Scientific Expedition and Research Program (STEP), Grant no. 2019QZKK0602, the National Natural Science Foundation of China under grants 41521004, 41991231, and 41975075, the Foundation of Key Laboratory for Semi-Arid Climate Change of the Ministry of Education in Lanzhou University, the China 111 Project (no. B13045). We also acknowledge support by the U.S. Department of Energy (DOE) Office of Biological and Environmental Research (BER) as part of BER's Subsurface Biogeochemistry Research Program (SBR) at the Pacific
Northwest National Laboratory (PNNL). PNNL is operated by Battelle Memorial Institute for the U.S. DOE under Contract Number DE-AC05-76RL01830. H. Liu acknowledges support by National Science Foundation (NSF-AGS-1419614 and NSF-AGS-1853050).

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
