# Peer review of "Technical note: Uncertainties in eddy covariance CO2 fluxes in a semiarid sagebrush ecosystem caused by gap-filling approaches"

_Atmospheric Chemistry and Physics, 2021_

## Author Comment (AC1)

**General Response**

We would like to thank the editor and two anonymous referees for their insightful comments and detailed suggestions. Following these suggestions, we revised the manuscript thoroughly to address all the comments. While the main points, the fundamental aspects of the presentation, and conclusions remain little changed based on the referees' comments, these comments did provide needed clarity in several sections of the manuscript.
* * *
**Anonymous Referee #1**

The study by Yao et al. tested and evaluated a suite of machine-learning algorithms in filling the data gaps of eddy-covariance $CO_2$ fluxes at a sagebrush site. They claimed that artificial neural networks and random forest algorithms perform better than the k-nearest neighbors and support vector machine algorithms. Last, they proposed a two-layer framework based on random forest algorithms and suggested providing a more reliable and robust alternative when filling extremely long data gaps.

The research topic is essential and attracts much attention in the science community. The manuscript is generally well-structured and written. I think the manuscript can be considered for publication in Atmospheric and Chemistry and Physics, after addressing a few general and specific comments.

***Answer:*** Thank you for the comments and recommendation!

[1] There have been studies, including several cited in the current manuscript, that tested and explored the applications of machine-learning algorithms in filling the data gaps of eddy covariance measurements. I suggest the authors summarize the previous studies' findings and highlight this study's uniqueness or innovative aspect (e.g., dryland ecosystem or the proposed two-layer approach). Below are two recent relevant studies:

Mahabbati, A., J. Beringer, M. Leopold, I. McHugh, J. Cleverly, P. Isaac, and A. Izady (2021), A comparison of gap-filling algorithms for eddy covariance fluxes and their drivers, Geosci. Instrum. Method. Data Syst., 10(1), 123-140, doi:10.5194/gi-10-123-2021.

Irvin, J., et al. (2021), Gap-filling eddy covariance methane fluxes: Comparison of machine learning model predictions and uncertainties at FLUXNET-CH4 wetlands, Agric For Meteorol, 308-309, 108528, doi: 10.1016/j.agrformet.2021.108528.

***Answer:*** Thank you for the suggestions! We added a brief review of the findings of the previous studies and emphasized the innovative aspect of this study in the introduction section of the revised manuscript.

[2] Presentation of technical details: Certain parts of technical information are not clearly explained or only presented later in the Result and Discussion sections. I suggest reorganizing

the texts and moving technical parts forward to the Materials and Methods section when feasible. It will also improve the readability by adding an overview subsection in the M&M, summarizing the study design and entire workflow.

***Answer:*** Many thanks for the suggestions! In the revised manuscript, we reorganized the texts and moved the technical information forward to the Materials and Methods section. In addition, we added an overview subsection in the Materials and Methods section to summarize the study design and entire workflow.

Specific Comments:

[3] Line 38: Since MDS is specifically called out, the original paper (Reichstein 2005) should be cited here.

***Answer:*** We cited the original paper.

[4] Line 42: MDS also may be the most widely applied method for gap-filling eddy covariance data and has been used extensively in FLUXNET data (e.g., Pastorello et al. 2020).

Pastorello, G., et al. (2020), The FLUXNET2015 dataset and the ONEFlux processing pipeline for eddy covariance data, Scientific Data, 7(1), 225, doi:10.1038/s41597-020-0534-3.

***Answer:*** We added the citation.

[5] Line 50-52: The sensitivity of dryland ecosystems to water availability is essential and maybe less addressed in previous gap-filling studies. This could be a unique contribution of this study. Yet, it is unclear to me whether and how this current study addresses this knowledge gap. Soil moisture or groundwater table seem not used as input variables. I'd suggest the authors considering exploring additional input variables for water availability.

***Answer:*** Thank you for insight suggestion! We included the precipitation as input variables but not the soil moisture and groundwater table because 1) the soil moisture measurements at these two adjacent stations had some issues for the study period and could not be used to fill the long gaps in tower data, and 2) the station is primarily fed by rainfall without groundwater access (Missik et al., 2019, 2021). In addition, using the available data after excluding the soil moisture gaps, our tests suggested that there was only a slight improvement in model performance for random forest with including the soil moisture as input variables (RMSE: 0.477 vs. 0.479; $R^2$: 0.762 vs. 0.760). RMSE is the root mean square error, and $R^2$ is the coefficient of determination. Therefore, including the precipitation as input variables should be good to represent water availability for this site. Also, NDVI and EVI can also reflect the variation of soil moisture to some extent.

[6] Line 78-79: Certain variables (e.g., soil) may be spatially varied among the stations. Consider briefly explaining whether or how the spatial bias is corrected.

*__Answer:__* Great points!  We agree with the referee's comment that certain variables may be spatially varied among the stations and lead to a biased relationship between the targeted fluxes and input variables. This would induce uncertainties to the gap-filled fluxes. In the revised manuscript, we added a short discussion of the potential impacts of spatial variability on the gap-filled fluxes (lines 58-63), but more attentions need to be paid to investigate this issue in future studies.

[7] Line 96-97: I think 10-fold cross-validation already implies resampling and grouping data for model training and validation. It doesn't need to state "repeated ten times". Would you please clarify it?

*__Answer:__* "10-fold cross validation" means randomly partition the original sample into 10 subsamples (9 of the subsamples for training the model and one for validating the model), and repeat the training and validating process 10 times, with each of the 10 subsamples used once as the validation data.  "Repeated ten times" refers to repeat the above 10-fold cross-validation procedure 10 times, which is 100 times (10x10) of training and validating processes. In this study, we used the 10-fold cross validation repeated ten times to improve the accuracy.

[8] Section 2.5: It may also be informative to explore the relative importance of input variables. For example, random forest allows the calculation of the relative importance of input variables, and such a feature has been utilized in previous studies to help interpret the results (e.g., Irvin et al. 2021). Other metrics have also been proposed to explain the variable importance, e.g., Knox et al. 2016; Kim et al. 2020.

Knox, S. H., J. H. Matthes, C. Sturtevant, P. Y. Oikawa, J. Verfaillie, and D. Baldocchi (2016), Biophysical controls on interannual variability in ecosystem-scale CO2 and CH4 exchange in a California rice paddy, Journal of Geophysical Research: Biogeosciences, 121, 978-1001, doi: 10.1002/2015JG003247.

Kim, Y., M. S. Johnson, S. H. Knox, T. A. Black, H. J. Dalmagro, M. Kang, J. Kim, and D. Baldocchi (2020), Gap-filling approaches for eddy covariance methane fluxes: A comparison of three machine learning algorithms and a traditional method with principal component analysis, Global change Biol, doi:10.1111/gcb.14845.

*__Answer:__* Thanks for your suggestion! Using the random forest, we estimated the relative importance of the input variables for three different scenarios: 1) all data without separating the daytime and nighttime periods, 2) daytime data, and 3) nighttime data, respectively (new Fig. 1).

[9] Line 135-141: Some technical details need to be explained here. (1) For "10% of the total data length", does it mean that an additional 10% of gaps (i.e., the total number of missing points) are created, or does it mean that artificial gaps are applied to 10% of the data records (i.e., some data points already missing)? I think it's likely the latter case since it's impossible to locate two months without any missing point. Would you please clarify it? (2) I assume the performance evaluation is done based on comparing score-0 observed data and estimated

values. Following the previous comment, what are the actual number of data points that are used for each comparison? Would unequal sample sizes affect the performance evaluation or statistic metrics used?

*Answer:* Many thanks for the insight comments! In the revised manuscript, we clarified the technical details (section 2.6). **(1)** The artificial gaps accounted approximately 10~12% of the NEE data with the score of 0, like those in previous studies (e.g., Moffat et al., 2007; Kim et al., 2020). We agree with the referee's comments that, for the NEE data after quality control, it is almost impossible to locate even couple days of data without any missing points. Thus, to generate a time series with artificial gaps, we randomly insert gaps with a certain length (e.g., one week) to the original time series (need to make sure that there is no overlap between the artificial gaps), and the above procedure stops when the number of the removed data points is just greater than 10% of the total number of score-0 data. **(2)** Therefore, for each artificial gap scenario, we had at least 10% of the score-0 observed data to be used to compare with the estimated values. In addition, each gap length scenario was permuted 10 times, resulting in 10 distinct time series for each gap length scenario, which allow us to examine mean performance of different ML algorithms on filling each gap length scenario, and reduce the potential effect of unequal sample sizes and gap positions on the performance evaluation or statistic metrics used. The mean (and standard deviation) of the actual numbers of data point were 3065 (1), 3085 (11), 3201 (90), and 3473 (221) for the gap lengths of one hour, one day, one week, and two months, respectively, accounting for approximately 10~12% of the score-0 NEE data.

[10] Line 141-143: Some of these metrics seem redundant. Several previous studies used Taylor diagrams, which may be considered, but I don't insist on it.

*Answer:* In the revised manuscript, we deleted the metric of absolute room mean square error, which had a similar variation with the relative room mean square error. We have also tried the Taylor diagrams, but it seems that the points are pretty close to each other and hard to show the variations.

[11] Line 157-163: As commented earlier, there may be spatial variability among the stations. Additional uncertainties may be introduced to flux gap-filling through these filled meteorological drivers. I suggest considering at least discuss the potential uncertainties.

*Answer:* Great points! Thank you for the suggestion! We added a short discussion about the potential uncertainties introduced to flux gap-filling through these filled meteorological drivers in the revised manuscript (lines 255-257; 403-405), but more attentions need to be paid to quantitatively investigate the potential bias in future studies.

[12] Figure 2: Please add units to the y-axis.

*Answer:* We added the units to the y-axis in Figure 2.

[13] Section 3.3: I think it's more accurate to call these estimated (or empirical) probability density functions since they are estimated based on data. I'd suggest being more specific about how they are calculated (in M&M). For example, the kernel density function may be the most commonly used. Also, there are more robust statistic tests for comparing density functions, e.g., Z-test.

*__Answer:__*  Thank you for the suggestion! We added a brief description of how the probability density function was calculated in Materials and Methods. It was calculated as the binned density distribution of measured (estimated) NEE divided by the bin width (0.2 $\mu mol\ m^{-2}\ s^{-1}$), and that's why it is not as smooth as a kernel density function. The purpose of comparing the probability density functions of measured and estimated data is to simply examine the performance of ML algorithms in terms of different ranges of NEE values, i.e., the regions where the frequencies of gap-filled data were underestimated or overestimated compared to the measured values. Therefore, we did not apply the kernel density function and Z-test.

[14] Section 3.4: I suggest briefly explaining and justifying the use of random measurement errors as a reference, e.g., Why? How to interpret it? In my opinion, gapfilling uncertainties are more like systematic errors, unlike random errors resulting from measurements or turbulence's stochastic nature. I'm not sure it's suitable to compare these two types of errors directly.

*__Answer:__*  Thank you for the suggestion!  Previous studies have suggested that the model errors of gap-filling algorithms should approach the measurement random errors of the EC method (Kim et al., 2020; Moffat et al., 2007; Richardson et al., 2008). Following these studies, we compared the model errors with the random measurement errors as a reference in this study (lines 213-216).

[15] Figure 5 and relevant texts: I think it needs a reference (e.g., pure observation or best-case prediction) for the performance evaluation here. I don't understand how the performance is evaluated here. Line 271-272 and analyses in Figure 6 seem to be a better option.

*__Answer:__*  Thanks for your suggestions! We reorganized the texts in the revised manuscript. With the 40 gap-filled NEE time series for each method, we first investigated the self-agreement of each method as the method with good self-agreement should have small variations in the accumulative NEE among different trails. And then we evaluated the model uncertainties in monthly NEE using the widely used MDS method as a reference.

[16] Line 291-293: I suggest providing a brief justification or discussion of the proposed approach.

*__Answer:__*  Thank you for the suggestion! In the revised manuscript, we moved the description of the proposed approach to the Materials and Methods, and added a brief discussion of the proposed approach in Section 4.2.
* * *
**Anonymous Referee #2**

Review report of "Technical note: Uncertainties in eddy covariance CO2 fluxes in a semiarid sagebrush ecosystem caused by gap-filling approaches" by Jingyu Yao, Zhongming Gao, Jianping Huang, Heping Liu, and Guoyin Wang

General Comments:

This study reports the result of applying different gap-filling approaches to access the change of NEE over the dry land ecosystems in the western US. Several types of artificial gaps were designed and put into the model to test the capability of using the various gapfilling approach. The authors noted that the performance among these available gap-filling approaches was silimiar, but all of these appraoches fail to fill large gaps over a period longer than two months. Among all selected gap-filling models, the ANN and RF approaches show a better model performance than others approaches, and the RF is relatively cheap in the cost of computational resources. The authors suggested that using RF to fill the gaps is the most efficient way to fill small gaps, such as the gap between hours and several days. In order to deal with the issue of data gaps over two months, the authors develop the strategy by adapting the information from the gap-filled dataset at a daily time scale. Again, the RF gap-filling approach was applied as defined as the second layer of the RF approach to avoid the issue of bias estimation (see Figure 2).

I agreed with the point of view for dealing with the EC gaps suggested by the authors, and the strategy of using a two-layer RF approach is robust as the results shown in this study. I think the idea proposed by the authors is good, but there is no need to stick to the RF approach. The same idea can also be applied to other gap-filling approaches to avoid bias estimation from long-term data gaps.

***Answer:*** Thanks for the suggestion! We added a brief discussion about the potential extension of the idea in the revised manuscript (lines 386-389).

As reported by the authors, the RF gapfilling approach shows a relatively good and stable result for filling short-term data gaps. Readers may be interested in understanding the importance of the potential variables used in the RF model. How does the RF model deal with the problems of collinearity among these variables? I suggested the authors report this part of the information to readers in order to support the conclusion made by the authors.

***Answer:*** Thank you for the suggestions! In the revised manuscript, we estimated the relative importance of the input variables using the RF algorithm. Our results suggested that solar radiation is the most important variable for the RF model trained using all the data and the daytime data, while air temperature and relative humidity are the most important variables for the RF model trained using the nighttime data. As for the collinearity among the input variables, we tested the RF model performance by excluding certain input variables. Our results indicated that the RF model generally has the best performance with all the input variables, although the

changes in model performance are very small after removing certain variables. For example, by excluding soil temperature, the $R^2$ only decreased about 0.02 (i.e., 0.759 vs. 0.761), while root mean square error (RMSE) only increased about 0.02 (i.e., 0.480 vs. 0.478). Our tests suggested that the RF model can deal with the collinearity of the input variables but may be more time efficient if using a smaller number of inputs. To be consistent with previous studies (e.g., Kim et al., 2020; Moffat et al., 2007), we also used all available meteorological variables as the inputs to train and evaluate the models (lines 179-189).

Besides, the structure of this manuscript is a bit confusing. Therefore, the idea of using two layer model can move to the section of methodology.

***Answer:*** Thank you for the suggestion! We moved the two-layer model to the Materials and Methods.

The issue of bias estimation from long-term data gaps should also be emphasized in the introduction section.

***Answer:*** Thank you for the suggestions! We emphasized the issue of bias estimation from long gaps in the introduction section (lines 55-58).

Base on the evaluation mentioned above, I support the publication of this manuscript as a technical note in ACP.

***Answer:*** Thank you for your suggestions and recommendation!

Technical Comments:

L19: …with this framework, the model performance is improved significantly, especially for the nighttime data.

Shall we separate the dataset into daytime and nighttime because the mechanism of producing the CO2 are quite different both for daytime and nighttime?

***Answer:*** Great points! Considering the difference in biological processes and meteorological conditions for daytime and nighttime, we evaluated the model performance using the daytime ("Daytime") and nighttime ("Nighttime") data separately, and the daytime and nighttime data together ("All"). Our results show that, when using the ANN and RF, the model performances are similar for "All" and "Daytime" data, though the model performances for "Nighttime" data are relatively poor.

L50: The motivation of this gap-filling practice was driven by the fact that dryland ecosystems are very sensitive to water availability,….

A short review of the global coverage of the dryland ecosystem is suggested. Readers may have an overall view of the importance of the dryland ecosystem under the current pace of global warming. Is the ecosystem going to be enlarged or reduced?

*Answer:* We added a short review of the global coverage of the dryland ecosystem and changing trends under climate change: "Dryland ecosystems, comprising around 40% of the Earth's land surface, play a critical role in determining the trend and interannual variability of the global terrestrial carbon budgets (Ahlström et al., 2015; Missik et al., 2021; Yao et al., 2020), though the expansion of projected global drylands under climate change is still under debate (Berg and McColl, 2021; Feng and Fu, 2013; Huang et al., 2015; Yao et al., 2020)" (lines 63-68).

L66: …197mm, …

How about the precipitation during the wet years and dry years?

*Answer:* According to the previous study (Justine et al., 2019), the annual precipitation at the site varied between 100 and 300 mm for dry and wet years, with an average of 197 mm for the period from 1986 to 2015. Annual precipitation in these four years were 217, 242, 169, and 210 mm, respectively (lines 95; 98-99).

L74: These data were sampled at a rate of 1 Hz…..

The frequency of 1Hz is too low for applying the eddy covariance approach to determine the surface exchange for the grassland ecosystem. Therefore, a 10Hz sampling rate is usually applied to determining the eddy flux for the surface exchange. I hope this is simply due to a typo. However, if the 1Hz is the actual system acquisition rate, I recommend conducting a spectrum analysis to be examined the contribution of high frequency to the total eddy flux by a theoretical correction.

*Answer:* The eddy covariance data were sampled at a rate of 10 Hz, and the meteorological data, such as four-component radiation, air temperature and relative humidity, were sampled at 1 Hz. We clarified the statement (line 102).

L90: … a score of 2 ….

I have no idea about the score. How to define the score in processing the EC data?

*Answer:* The difference between score "1" and "2" is that the meteorological data are available from the tower measurements for score "1"; while the difference between score "0" and "1" is that the flux data are quality assured for score "0". We clarified this in the revised manuscript (123-125).

L104: … we use two hidden layers with 12 and 10 nodes in the first and second hidden layers…

Why 10 and 12 layers. Any literature to support these values?

*Answer:* Previous studies (i.e., Dengel et al., 2013; Jammet et al., 2015; Kim et al., 2020; Knox et al., 2016) have suggested to use two hidden layers for the ANN but did not specify the nodes. Here, we used the "caret" R package (Kuhn et al., 2020) with a 10-fold cross-validation repeated ten times to optimize the required parameters in each algorithm.

L111: Here, the optimized k value is 9...

Again, how to determine this value?

***Answer:*** The value of k is optimized using the "caret" R package (Kuhn et al., 2020) with a 10-fold cross-validation repeated ten times.

L265: .... RFs denotes the proposed two-layer RF based gap-filling framework, and MDS is the marginal distribution sampling algorithm....

The abbreviation of RF makes readers confused. Therefore, I suggested using RF-2L to represent the two-layer random forest approach.

***Answer:*** Great! We replaced the abbreviation of "RFs" with "RF-2L" in the revised manuscript.

L287: Section: A two-layer RF based gap-filling framework for extremely long gaps

I was confused about the two-layer RF approach while reading the manuscript for the first time. Did you mean that combining the approaches from two layers is for ANN + RF, but there is no information on how to combine these two approaches? Therefore, I suggested moving this section to methodology.

***Answer:*** Thanks for the suggestion! We have moved the section to the Materials and methods.

L309-L310: The RF algorithm outperforms the other algorithms in terms of the overall performance. I would like to see the importance of the potential variables that were applied in this study. How to deal with this issue of the collinearity problem?

***Answer:*** Thanks for the comments! Please see our answer above.

**Reference**

[revised manuscript text omitted]